# Development of a Stress Sensor for In-Situ High-Pressure Deformation Experiments Using Radial X-Ray Diffraction

**Jennifer Girard [1],\*, Reynold E. Silber [1] , Anwar Mohiuddin [1,2] , Haiyan Chen [3] and Shun-ichiro Karato [1]**

[1]  Department of Geology and Geophysics, Yale University, New Haven 06511 CT, USA;
   reynold.silber@yale.edu (R.E.S.); anwar.mohiuddin@yale.edu (A.M.); shun-ichiro.karato@yale.edu (S.-i.K.)
[2]  Intel Ronler Acres Campus, 2501 NE Century Boulevard, Hillsboro 97124 OR, USA
[3]  Mineral Physics Institute, Stony Brook University, Stony Brook 11794 NY, USA; haiyan.chen@stonybrook.edu
\*  Correspondence: jennifer.girard@yale.edu

**Abstract:** We developed a stress sensor for in-situ deformation experiments using synchrotron radial X-ray diffraction. This stress sensor provided nearly diffraction-plane-independent stress that, when used in series with a sample, reduced the uncertainty of the average stress estimation acting on a sample. Here, we present the results of a study where pyrope was used as a stress sensor. Using a Deformation-DIA (D-DIA) high-pressure deformation apparatus, pyrope, olivine and alumina were deformed in the same run/cell assembly placed in series along the compression direction. Deformation experiments were conducted at pressures between 4 and 5 GPa and temperatures between 730 and 1273 K with strain-rates between $10^{-5}$ and $10^{-6}$ s$^{-1}$. Stresses estimated from various (hkl) planes in pyrope were nearly the same; i.e., pyrope is plastically isotropic with ≤10 % variation with (hkl). However, stresses from various (hkl) planes in olivine and alumina varied by approximately a factor of 3. Comparisons between average stresses inferred from pyrope and those from different diffraction planes in olivine and alumina showed that the average stress in these materials evolved from low-end stress, estimated from various (hkl) planes at small strain, to high-end stress at a large strain. This suggests that the rate-controlling slip system in these materials changes from the soft to the hard slip system with strain.

**Keywords:** high pressure; deformation; in-situ; D-DIA; stress sensor

## 1. Introduction

In a classic deformation experiment, one measures macroscopic strain and stress using a displacement transducer and a load cell [1,2]. From these measurements, one obtains a relationship between macroscopic stress and strain. The results provide key data sets to help us understand deformation of a material.

A load cell cannot be used under high-pressure conditions because a sample is surrounded by various materials and, thus, the load (or the torque), determined outside of a sample assembly, does not reflect directly the stress acting on a sample. Radial X-ray diffraction is a powerful tool for estimating the stress in a sample during plastic deformation, including deformation under high pressures (see e.g., [3–5]). This is possible using high-intensity X-rays generated by a synchrotron radiation facility. In this approach, diffracted X-rays from a sample are collected, at different orientations with respect to the macroscopic stress orientation, to estimate the stress. Strain (and strain rate) is estimated by collecting X-ray radiography images of the sample during deformation. A relationship between stress and strain is obtained and is essential to understanding the plastic properties of a material.

However, a fundamental challenge in this method of characterization of plastic properties is the fact that the relationship between the macroscopic stress and the stress estimated from X-ray diffraction is not simple. In this method, strain is measured by the displacement of the sample (i.e., macroscopic strain), but macroscopic stress (the average stress acting on a sample) is estimated from microscopic strain (the strain in individual grains) determined by X-ray diffraction. In this technique (called "radial X-ray diffraction"), one measures the variation in *d*-spacing of various lattice planes, $\frac{\Delta d_{hkl}}{d_{hkl}}$, and converts it to stress using the known elastic constants of a sample (see e.g., [6–8]). However, it is often noted that the estimated stress from different diffraction planes of a material differs substantially (sometimes by more than a factor of 2–3 [9,10]), and it is unclear how one can estimate the average (macroscopic) stress acting on a sample from a variety of stress values estimated from radial X-ray diffraction. Karato [11] developed a theory to explain this by the large variation in stress from different diffraction planes. Karato's theory shows that a large variation in stress is largely due to plastic anisotropy that affects the distribution of local stress in an aggregate. Plastic anisotropy is usually much larger than elastic anisotropy and can lead to a large variation in the estimated stress from lattice strain ($\frac{\Delta d_{hkl}}{d_{hkl}}$).

The purpose of this work is to develop an isotropic stress sensor from which the average stress acting on a sample is estimated with a smaller uncertainty than that estimated with previous stress sensors, and to test it under high-pressure and temperature conditions using olivine and alumina as samples.

In some previous studies, a stress sensor was used to estimate the stress acting on a sample when the sample itself cannot be used to estimate the stress during deformation (e.g., single-crystal deformation) (see e.g., [12–16]). In these cases, polycrystalline alumina was used as a stress sensor. However, these stress sensors have large plastic anisotropy, which led to large uncertainty in the stress estimation.

## 2. Materials and Methods

### 2.1. Sample Preparation and Analysis

The theoretical guideline for choosing a material for a stress sensor is simple: the (hkl) dependence of estimated stress from the radial X-ray diffraction comes from elastic and plastic anisotropy of a material [11]. Therefore, a good material as a stress sensor is a material with small elastic and plastic anisotropy. Most garnets are elastically almost isotropic (see e.g., [17]). Additionally, due to the high symmetry of slip systems, garnets are also plastically nearly isotropic (see e.g., [18,19]). Consequently, we chose an end-member of the garnet group with formula $Mg_3Al_2Si_3O_{12}$ (pyrope) as a sensor material.

Pyrope ($Mg_3Al_2Si_3O_{12}$) garnet was synthesized from an oxide mixture ($MgO$–$Al_2O_3$–$SiO_2$). The oxides were thoroughly mixed in an agate mortar and a ball roller mixer for 12 h. The mixture was then melted at the temperature (*T*) of 1973 K in a vertical tube furnace (Deltech Inc. manufactured, Denver, CO, USA), following the melting phase diagram proposed by Irifune and Ohtani [20]. The molten mixture was quenched by dropping it in distilled water, resulting in pyrope glass, which was subsequently ground and analyzed by X-ray diffraction (Rigaku miniflex 600, manufactured, Tokyo, Japan) to ensure the absence of unwanted crystalline phases due to inhomogeneous melting. The glass powder was hot pressed in a 25/17 cell assembly at the pressure (*P*) of ~3 GPa and 1573 K for 1 h to form fine grain polycrystalline pyrope following Irifune et al. [21] using a Kawaii-type large volume high-pressure apparatus (Try Engineering, Japan), available at Yale University.

San Carlos olivine crystals (San Carlos NM, CA, USA) (($Mg_{0.91}$, $Fe_{0.09}$)$_2SiO_4$), also referred to as Fo91, without any visible inclusions, were handpicked and ground to a powder using a hand-operated hydraulic piston press and then a ball mill. The powder was sorted to obtain fine-grained olivine with grain size less than ~1 μm using a sedimentation technique. The fine-grained powder was hot pressed in a 25/17 cell assembly at 3 GPa and 1073 K for 1 h. Recovered samples were polished and analyzed using a SEM (Scanning Electron Microscope XL30 ESEM-FEG, FEI, Hillsboro OR, USA) to confirm that the grain size was ≤ 1 μm.

Each hot-pressed sample was then sliced to the appropriate thickness (0.2–0.4 mm for pyrope and 1.3 mm for olivine; alumina is part of the cell assembly in D-DIA experiments), and core-drilled into 1 mm diameter cylinders for D-DIA uniaxial deformation experiments.

## 2.2. D-DIA High-Pressure Deformation Experiments

We performed two high-pressure deformation experiments using a D-DIA apparatus at the 6-BM-B white X-ray beamline at Advanced Photon Source (APS) in Argonne National Laboratory. These are referred to by the run number throughout the text, and also shown this way in Table 1 in the results section (i.e., San 430 and San 452).

Pressure was increased to a desired value by applying a load on all of the six anvils. Heating was usually achieved by using a graphite heater as shown in Figure 1A (used in run San 430). To minimize the temperature gradient, we used a stepped heater as shown in Figure 1B (used in run San 452). In addition, two lateral thermocouples were placed; one on top and one at the center of the cell, to monitor the possible temperature gradient in the cell assembly.

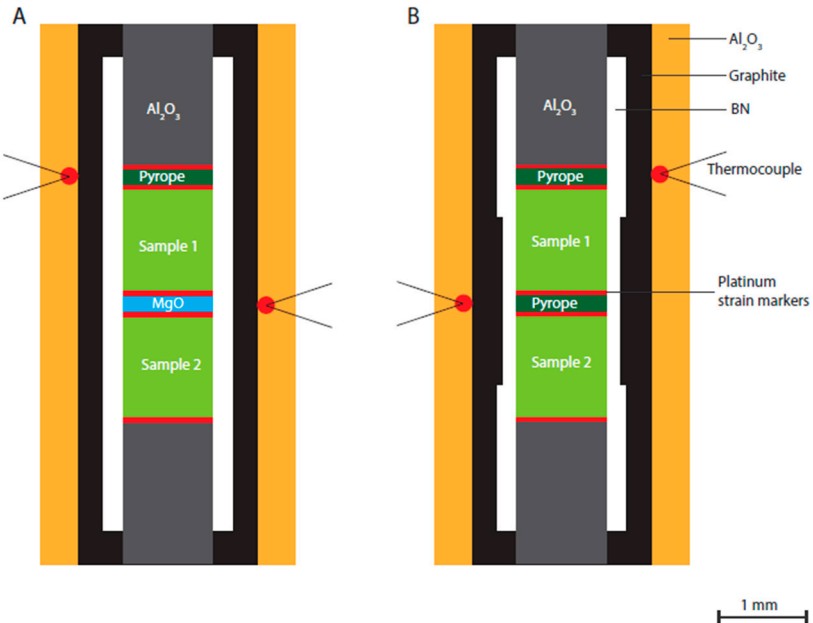

**Figure 1.** (**A**) Schematic of the traditional D-DIA cell assembly used in San 430 to test the pyrope stress sensor and synthesized fine grain olivine samples. (**B**) A D-DIA cell with a graphite stepped furnace developed to minimize the temperature gradient in the sample in San 452.

During the experiments, temperatures were estimated from thermocouple readings (for run San 430), and from a temperature calibration run (for the run San 452). A temperature calibration run was performed with thermocouples positioned at the center of the cell assembly, using NaCl as a sample at $P = 5$ GPa (the pressure was estimated from NaCl's equation of state). The pressure and temperature were also estimated using the equation of state of MgO and platinum (for run San 430), and using the equation of state of pyrope and platinum (for run San 452).

To deform samples, the top anvil was advanced relative to the bottom anvil under a nearly constant pressure. These samples were surrounded by materials that were X-ray transparent and thus stress and strain could be monitored during deformation. A schematic of our cell design is shown in Figure 1. The cell consisted of two machinable alumina pistons used to compress the samples (olivine stress sensor and pyrope stress sensor). The samples were sandwiched between platinum foils used as strain markers.

**Table 1.** Summary of deformation conditions for runs San 430 and San 452.

| Run # | *P* (GPa) | *T* (K) | Stress (GPa) [†] | | | Strain Rate (s⁻¹) [‡] | | Total Strain (%) [‡] |
|---|---|---|---|---|---|---|---|---|
| | | | Pyrope | San Carlos Olivine | Alumina | Pyrope | San Carlos Olivine | |
| San 430 | 4.4 ± 1 * | 1273 ± 130 * | 0.16 ± 0.01 | 0.12 ± 0.04 | 0.14 ± 0.04 | $0.7 \pm 0.1 \times 10^{-6}$ | $3.5 \pm 0.4 \times 10^{-6}$ | 14 ± 1.5 |
| | | | 0.27 ± 0.02 | 0.17 ± 0.06 | 0.230 ± 0.06 | $2 \pm 0.2 \times 10^{-6}$ | $7.9 \pm 0.8 \times 10^{-6}$ | |
| San 452 | 4.8 ± 0.6 ** | 730 ± 90 ** | 3.6 ± 0.1 | 3.04 ± 0.7 | 3.05 ± 0.4 | - | $3.7 \pm 0.4 \times 10^{-6}$ | 5 ± 0.6 |

* Obtained from the EOS (Equation of State) of Pt and MgO and confirmed with a thermocouple during the experiment (a 10 K difference between the thermocouple and the EOS estimate.). The uncertainty in *P-T* was calculated using the uncertainty on d-spacing resulting from peak fitting from platinum and from MgO. ** Temperature and pressure reported in the table are the average values with one standard deviation calculated from a power versus temperature calibration run and from the EOS of pyrope and platinum. [†] Stress values reported in the table are weighted average stress values at steady state condition (constant stress, constant strain rate). The error reported corresponds to one standard deviation of stress estimated for each (hkl) peak. [‡] Strain and strain rate uncertainties were calculated using the sample length measurement uncertainty (10%). Strain of pyrope in San 452 was too small to be measured accurately; therefore, strain rate could not be estimated.

Strain in each sample was measured by X-ray absorption images of strain markers collected periodically during a deformation experiment. The distance between strain markers was measured from the images at various times to estimate the strain rate [22].

Stress was estimated from energy dispersive X-ray diffraction, with a constant 2θ collected through a conical slit [23] using a set of 10 detectors along 10 different azimuthal angles (0°, 22.5°, 45°, 67.5°, 90°, 112.5°, 135°, 157.5°, 180°, and 270°).

The diffracted X-ray signals collected at each azimuthal angle were used to estimate the lattice strain which, was then used to calculate the differential stress on crystals during deformation, following Singh [8], which connects lattice strain to stress (Equation (1)).

$$d_{hkl}(\psi) \propto d^0_{hkl}\left\{1 + \frac{\sigma_u}{6M}\left(1 - 3cos^2\psi\right)\right\} \tag{1}$$

where $d_{hkl}$ is the *d*-spacing for the lattice plane (hkl), $d^0_{hkl}$ is the *d*-spacing in hydrostatic conditions for the lattice plane (hkl), $M$ is the elastic modulus of the crystallographic orientation, $\psi$ is the azimuthal angle, and $\sigma_u$ is the uniaxial stress applied on the material. Note that Singh's theory [8] ignores the influence of plastic anisotropy, and, consequently, stress estimated from different diffraction planes (hkl) shows a large variation when plastic anisotropy is strong.

X-rays diffracted from the olivine sample, as well as the pyrope stress sensor and alumina piston, were collected throughout the deformation experiment (Figure 2), and stress was calculated using Equation (1) with the appropriate elastic constants [24–26]. An example of measured lattice strain (i.e., *d*-spacing) for different (hkl) peaks for pyrope, olivine and alumina is shown in Figure 3.

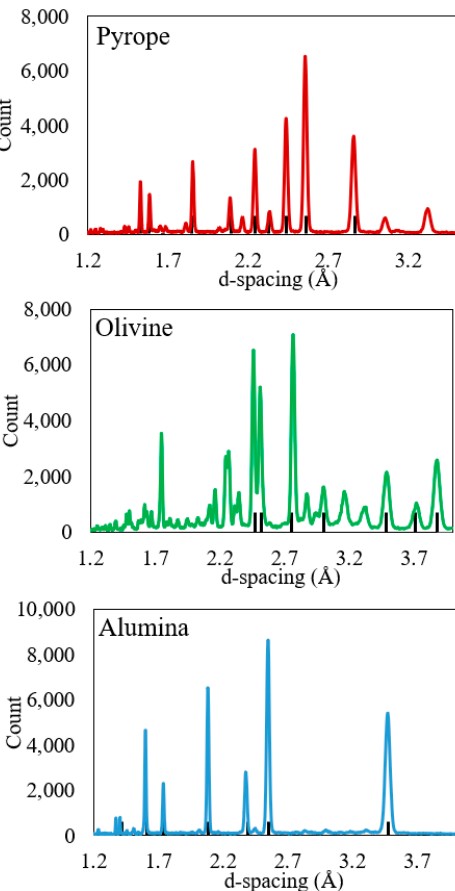

**Figure 2.** Example of the Energy dispersive X-ray diffraction spectra collected on each sample (pyrope, olivine and alumina) from run San 430 collected before the experiment, at room pressure and room temperature. The black lines indicate the diffraction peaks analyzed for stress calculation.

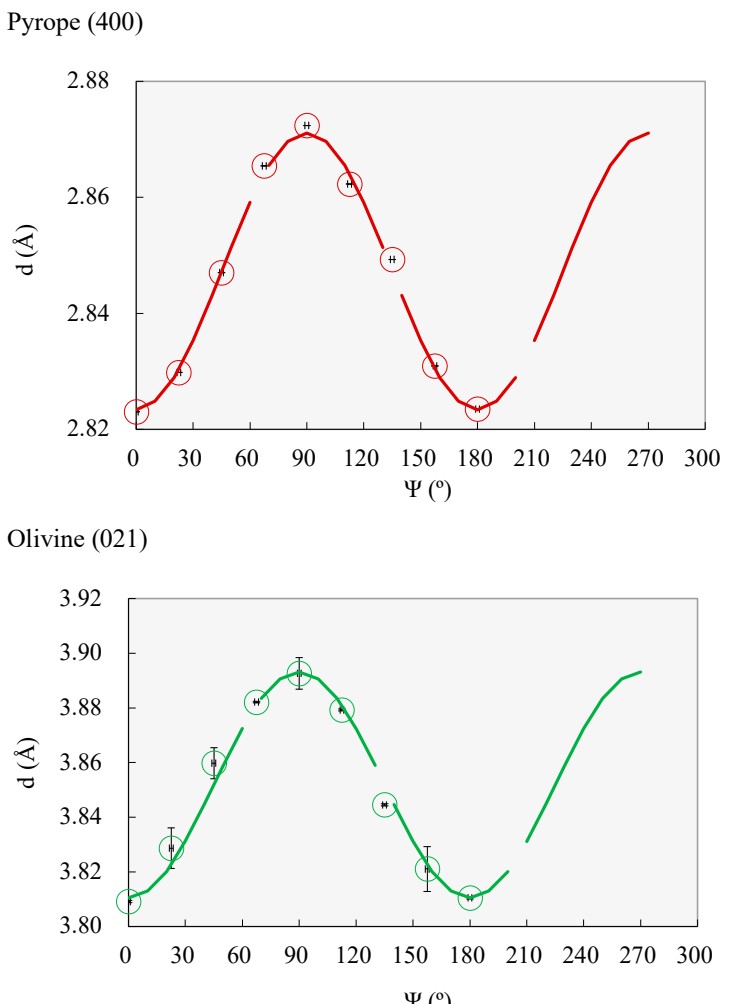

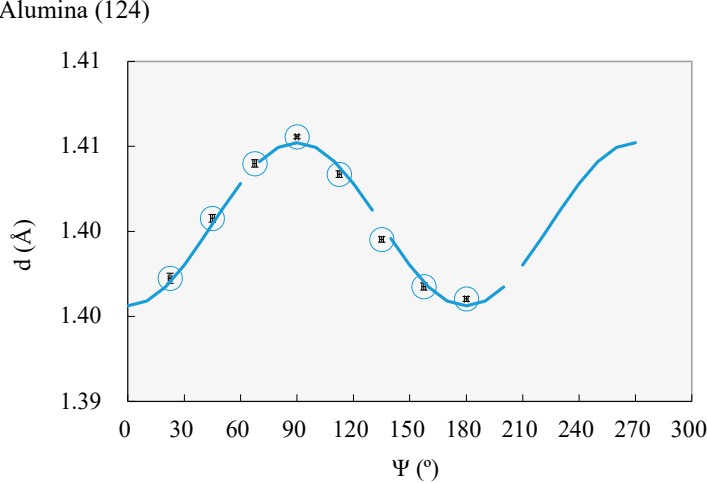

**Figure 3.** Example *d*-spacing versus azimuthal angle (ψ) for different (hkl) diffraction peaks used to estimate stresses for pyrope (400), Olivine (021) and alumina (124).

In our cell assembly (Figure 1), two samples as well as a stress sensor were stacked (+ alumina pistons). When friction along the edge of each sample was small, stress in each sample and in the stress sensor was the same. To evaluate the influence of friction, we used two stress sensors; one stress sensor

at the top of the sample chamber and a second stress sensor in the center for run San 452. Stress $\sigma_{hkl}$ for each (hkl) diffraction plane (for pyrope, alumina and olivine) is shown in Figure 4. Uncertainties of each individual $\sigma_{hkl}$ were calculated from the uncertainty in peak fitting. These uncertainties were used as weights in weighted average stresses calculations and are reported in Table 1; stress uncertainty in Table 1 corresponds to one standard deviation.

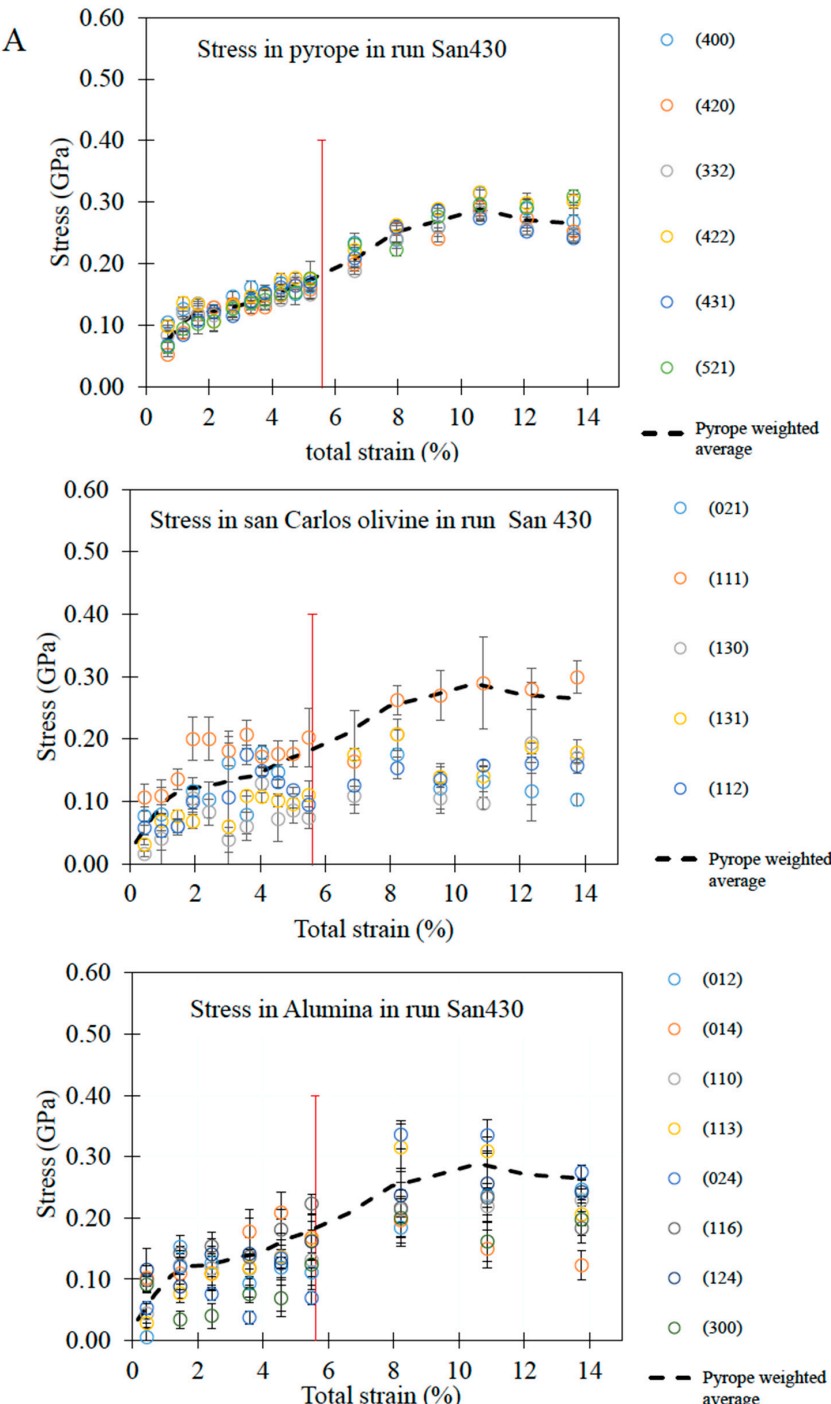

**Figure 4.** *Cont.*

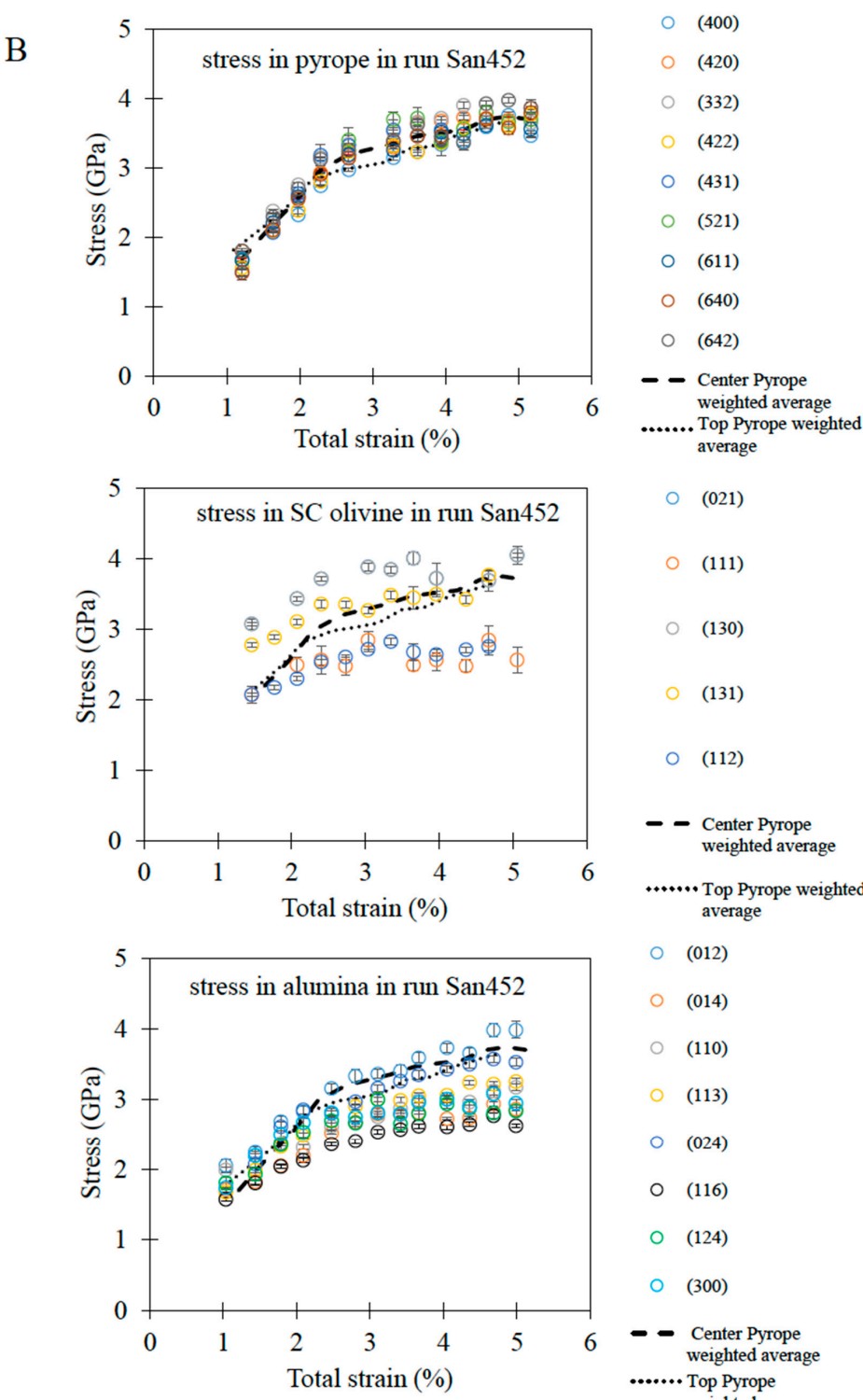

**Figure 4.** (**A**) Stress versus total strain for run San 430. The dashed line represents the weighted average stress estimate for pyrope. The red vertical line marks the time when the strain rate was increased. (**B**) Stress versus total strain for run San 452 estimated for various (hkl) diffraction peaks for pyrope, olivine and alumina.

## 3. Results and Discussion

Figure 4 shows the stress estimated from pyrope, olivine and alumina, collected in two different high-pressure uniaxial deformation experiments. One run (San 430; Figure 4a) is at P = 4.4 GPa under

low stress, and another run (San 452; Figure 4b) is at P = 4.8 GPa. Due mainly to the differences in temperature (500 K difference), the stress levels are markedly different between these two runs.

We report stresses $\sigma_{hkl}$ estimated from various (hkl) peaks from each sample with their respective uncertainties $\Delta\sigma_{hkl}$ obtained using peak fitting uncertainties. The dashed line represents the weighted average stress estimated from the pyrope stress sensor $\sigma_{(hkl)}$, where individual uncertainties $\Delta\sigma_{hkl}$ were used as weights. The red vertical line indicates the time at which the strain rate was changed. In San 430, a pyrope stress sensor was placed only at the top of the column, but in San 452, we placed two pyrope stress sensors (one on top of the deformation column and one in the center of the cell assembly). Figure 4b shows stress estimated from the pyrope at the center of the cell; the dotted line is the weighted average stress estimated from the pyrope's position at the top. Both stress estimates are similar and thus friction can be considered negligible.

We note that, because of its isotropic properties, variation in stress among different diffraction planes for pyrope was small with a maximum of 10% whereas, for olivine and alumina, stress values varied substantially with the diffraction planes (up to a factor of 3 or more).

Therefore, pyrope works as a good sensor to estimate the average stress acting on a sample. Consequently, a comparison of average stress estimated from the stress sensor and the stresses estimated from various diffraction planes in olivine provides a new insight into the role of various slip systems in deformation of a polycrystalline aggregate (see e.g., [27–29]). Our results show that, in these two runs, the average stress estimated from various diffraction planes in pyrope (represented by the dotted and dash lines) was in the upper part of the stress values estimated from different diffraction planes in olivine and alumina. This means that the strength of alumina and olivine aggregates are largely determined by their respective strong slip system(s) (since the largest stress is required to deform the aggregate), and therefore implies that deformation of each aggregate in these cases occurs at nearly homogeneous strain.

Finally, we compared the stress estimated in alumina in our study with Raterron et al. [30]. We confirm that the ordering of the $\sigma_{hkl}$ is the same in both studies (e.g., stress estimated from the (012) diffraction plane is always the largest and the (116) diffraction plane always shows the smallest stress). However, in Raterron et al. [30], large anisotropy in stress estimated from alumina prevented an accurate stress estimation. Thus, they used EPSC (Elastic Plastic Self Consistent) modeling to calculate "true" stress. Their results showed that, at the end of deformation, EPSC modeling agreed with the lower end of stresses estimated in alumina. This is not consistent with our observations using a pyrope stress sensor. Since these two methods differ, it is difficult to explain where the variances come from. In our study, stress in pyrope was directly measured from X-ray diffraction but the "true" stress corresponding to the EPSC modeling was calculated following some assumptions. Therefore, some of the EPSC assumptions may need to be refined.

We also note that the average stress of a deforming material evolves from low end to high end with strain, suggesting that the slip system that controls the strength of an aggregate changes from the soft slip system at a small strain to the hard slip system at a large strain (this is clearly seen for run San 452 where the relative error in stress estimate is small compared to run San 430). This is consistent with an idea presented by Karato [31]. However, it is not clear if this is the case for more general cases. At higher strain, the stress–strain distribution may change. A stress sensor can be used to investigate the evolution of stress–strain distribution during plastic deformation. A good stress sensor material can be any material with low (or no) plastic anisotropy. This technique can also be applied to deformation of two-phase mixtures where the evolution of stress–strain with strain is an important issue in characterizing the processes of shear localization.

**Author Contributions:** J.G. and S.-i.K. conceived the idea of the study. J.G., A.M. and R.E.S. performed sample preparation and conducted experiments at Argonne National Laboratory with the help of H.C. J.G. analyzed the data and wrote the manuscript, which was edited by the co-authors. All authors have read and agreed to the published version of the manuscript.

**Funding:** This research was funded by National Science Foundation, NSF grant number EAR 1764271.

**Acknowledgments:** It is our pleasure to contribute to a special volume dedicated to Orson Anderson. Orson made a huge contribution to our community by introducing a rigorous physics approach to geological sciences. We hope that our work represents one of the contributions to follow the approach pioneered by Orson Anderson. The authors would like to thank COMPRES for partly supporting the facility 6-BM-B, where the data for this study were collected. We are also grateful to W. Samella and C. Fiederlein for helping with preparation of parts for 25/15 as well as the D-DIA cell assemblies used in this study.

**Conflicts of Interest:** The authors declare no conflict of interest.

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
