# Peer review of "Development of a Stress Sensor for In-Situ High-Pressure Deformation Experiments Using Radial X-Ray Diffraction"

_minerals, doi:10.3390/min10020166_

Round 1
Reviewer 1 Report
This is a very nice paper. It is in keeping with Orsen Anderson's interest in experimental details and precise science. This paper examines the measurement of stress with X-rays in a DDIA. The data from this machine, while giving great insight into the details of stress distribution within a sample, fell short in defining the load per unit area simply because the stress is in fact heterogeneous. But ultimately, flow laws need to know the load per unit area. Here the authors that garnet can serve as an in situ piezometer that can provide this valuable information. I should say that this paper convinces me that I should use pyrope in my experiments.
Should be published. No specific comments
Author Response
Response to Reviewer 1 Comments
Comments and Suggestions for Authors
This is a very nice paper. It is in keeping with Orsen Anderson's interest in experimental details and precise science. This paper examines the measurement of stress with X-rays in a DDIA. The data from this machine, while giving great insight into the details of stress distribution within a sample, fell short in defining the load per unit area simply because the stress is in fact heterogeneous. But ultimately, flow laws need to know the load per unit area. Here the authors that garnet can serve as an in-situ piezometer that can provide this valuable information. I should say that this paper convinces me that I should use pyrope in my experiments.
Should be published. No specific comments
We thank the reviewer for their review recognizing the importance of our contribution.

Reviewer 2 Report
The paper regards the development and testing of a new stress sensor with a low elastic and plastic anisotropy, in order to predict, with a minor possible uncertainty, the relationships between stress and strain in studies on high-pressure deformation.
The manuscript is well written but need to be improved clarifying some aspects in the Introduction and description of experiment. Some arguments, understood by the authors, are not sufficiently explained and should be made available also for the readers. For example, the difference between macroscopic and microscopic stress and strain should be explained. On the other hand, it is not immediately understandable that the material tested is not a monocrystal but microcrystalline. The status of pyrope before to apply stress should be described: some analyses were made to investigate how much and how it deviates from perfect crystal?
The manuscript needs also of the correction of some errors in the description of the material tested and the use of appropriate terms in mineralogical and crystallographic field should be improved. The material tested is a synthetic pyrope, an end member of a garnet group with a chemical formula Mg3Al2Si3O12. Some mention about its space group and its possible slip systems should be added. In the section 2. Materials and Methods in the description of the sample preparation, pyrope is defined as glass, but the x-ray diffraction (see Fig. 2) shows clearly that it shows a high crystallinity. The term glass is commonly used for amorphous material or material with very low crystallinity and so should be deleted.
In the following, some corrections are listed:
Line 14: the use of terms pyrope garnet should be replaced by: an end member of a garnet group with a chemical formula Mg3Al2Si3O12 (pyrope).
Line 30: after load cell, introduce some references, please.
Line 62: the sentence should be modified as: we chose an end-member of garnet group with formula Mg3Al2Si3O12 (pyrope).
Line 67 and line 68: please delete the term glass and use the term crystal or sample.
Line 71: the description of San Carlos olivines should be introduce and explained.
Line 122: The Figure 2 should be better explained.
Line 129: In the caption of Figure 3 after azimuthal angle the Greek letter (ψ) should be added. The choice of these reflections should be explained. In this Figure the d-spacing variations is of about 0.05 Å for the reflection 400 of Pyrope, 0.08 Å for the reflection 021 of Olivine and 0.01 Å for Al2O3. This evidence would show that the alumina exhibits a better isotropic behaviour…please explain.
Line 156-157: Figure 4 and 4b should be distinguished in Fig. 4 and Fig. 5, respectively, with two different figure captions.
Author Response
Response to Reviewer 2 Comments
Comments and Suggestions for Authors
The paper regards the development and testing of a new stress sensor with a low elastic and plastic anisotropy, in order to predict, with a minor possible uncertainty, the relationships between stress and strain in studies on high-pressure deformation.
Point 1: The manuscript is well written but need to be improved clarifying some aspects in the Introduction and description of experiment. Some arguments, understood by the authors, are not sufficiently explained and should be made available also for the readers. For example, the difference between macroscopic and microscopic stress and strain should be explained.
We thank the reviewer for their comments. More details were added to the introduction to ease the reader in understanding the difference between macroscopic stress and local stress in aggregate. (line 42-45)
Point 2: On the other hand, it is not immediately understandable that the material tested is not a monocrystal but microcrystalline. The status of pyrope before to apply stress should be described: some analyses were made to investigate how much and how it deviates from perfect crystal?
We thank the reviewer for pointing this out. We added more details in the main text (lines 80-81) for better understanding. All materials were polycrystalline samples. Pyrope was synthesized from oxide mixture that we melted to form a perfect glass. The glass was then hot-press at 3GPa and annealed at 1573 K for 1 h to form fine-grained polycrystalline pyrope following Irifune et al. [1]. The X-ray diffraction of polycrystalline pyrope collected at room pressure and room temperature before the experiment is presented in Figure 2.
Point 3: The manuscript needs also of the correction of some errors in the description of the material tested and the use of appropriate terms in mineralogical and crystallographic field should be improved. The material tested is a synthetic pyrope, an end member of a garnet group with a chemical formula Mg3Al2Si3O12. Some mention about its space group and its possible slip systems should be added. In the section 2. Materials and Methods in the description of the sample preparation, pyrope is defined as glass, but the x-ray diffraction (see Fig. 2) shows clearly that it shows a high crystallinity. The term glass is commonly used for amorphous material or material with very low crystallinity and so should be deleted.
We understand that Crystallographic information on pyrope might be important, however pyrope is not the only material that might be use as stress sensor. Any material with low (or no) plastic anisotropy can make a perfect stress sensor, (this was added 217) regardless of crystal system or space group. Therefore, we didn’t add this information since this is not what makes a good stress sensor.
We also added more information in material and methods regarding pyrope synthesis: We first synthesize pyrope glass, and by hot-pressing it changed to a polycrystalline aggregate. We described this process more clearly in lines 72.
In the following, some corrections are listed:
Point 4: Line 14: the use of terms pyrope garnet should be replaced by: an end member of a garnet group with a chemical formula Mg3Al2Si3O12 (pyrope).
We thank the reviewer for pointing this out and we made the appropriate correction in the text (line 71).
Point 5: Line 30: after load cell, introduce some references, please.
The relevant references are given in the manuscript now. We quote pioneers of high-pressure deformation Griggs 1936, Paterson 1970 as examples (line 30). Those first high pressure deformation experiments were carried in laboratory using load cell and transducer as described in their papers.
Point 6: Line 62: the sentence should be modified as: we chose an end-member of garnet group with formula Mg3Al2Si3O12 (pyrope).
We made the correction
Point 7: Line 67 and line 68: please delete the term glass and use the term crystal or sample.
We initially form glass with pyrope composition, and then synthesized fine-grained polycrystalline sample. To avoid possible confusion, we described this sample preparation procedure more clearly in the revised version.
Supporting Figure: X-ray diffraction collected showing the amorphous structure of the “pyrope glass”, which was hot pressed to polycrystalline pyrope (see X-ray diffraction in Fig. 2)
Point 8: Line 71: the description of San Carlos olivines should be introduce and explained.
We added information on composition “San Carlos olivine crystals ((Mg0.91,Fe0.09)2SiO4),” in line 82.
Point 9: Line 122: The Figure 2 should be better explained.
We clarified this in figure captions. “Example of energy dispersive X-ray diffraction spectra collected on each sample (pyrope, olivine and alumina) from run San 430 collected before the experiment, at room pressure and room temperature. The black lines indicate the diffraction peaks analyzed for stress calculation.”
Point 10: Line 129: In the caption of Figure 3 after azimuthal angle the Greek letter (ψ) should be added. The choice of these reflections should be explained. In this Figure the d-spacing variations is of about 0.05 Å for the reflection 400 of Pyrope, 0.08 Å for the reflection 021 of Olivine and 0.01 Å for Al2O3. This evidence would show that the alumina exhibits a better isotropic behavior…please explain.
We added the Greek letter (ψ) to the figure caption.
There are no reasons for the selection of (hkl) in Figure 3. We use these (hkl) to illustrate typical examples of diffraction peaks. The variation in d-spacing as a function of azimuthal angle (ψ) is used to estimate the stress following Singh et al (1998) relation (equation 1) that depends on elastic constant (different for different material). The magnitude of variation in d-spacing for different materials is largely dependent on the elastic moduli and has nothing to do with anisotropy.
Point 11: Line 156-157: Figure 4 and 4b should be distinguished in Fig. 4 and Fig. 5, respectively, with two different figure captions.
Figure 4a and Figure 4b show results from two different runs (Fig. 4a: run San430), Fig. 4b: run San452). These are similar runs with different temperatures and hence different stress levels. We believe that showing these two sets in a single Figure will be better to show the differences between the two.

Reviewer 3 Report
Review for "Development of a stress sensor for in-situ high-pressure deformation experiments using radial X-ray diffraction" by J. Girard, R.E. Silber, A. Mohiuddin, H. Chen, and S. Karato, submitted to Minerals
This papers presents results regarding high pressure deformation experiments. The authors present deformation results on pyrope and argue that, because of its elastic and plastic anisotropy, garnet is a good material to serve as a reference for stress evaluation in diffraction experiments.
The main conclusions are new, scientifically sound, supported by the data, and useful to the community. As such, this paper should be published. I do have questions and comments (see below). I hence recommend publication after corrections.
1- Using stress sensors in D-DIA experiments is not new. Several groups used the alumina piston to evaluate stress in olivine, for instance. Scientific honesty requires citations to the relevant literature.
2- Alumina as a stress sensor, how x-ray stresses could be corrected using EPSC models, and the effect of gradients in the D-DIA column were tested and published in Raterron et al Rev. Sci. Instrum. 2013. This paper should be compared to the present results and discussed.
3- Line 133: "When friction along the edge of each sample is small, stress in stress in each sample and in the stress sensor is same".
Actually, no. If 3 is the vertical direction. s33 is, in fact, the same in the sample as well as in the stress tensor. Stress, as defined in Eq. 1 however is not s33, but sU = s33-s11. The authors hence assume that s11 is the same in all materials.
Is s11 the same in all materials? I do not know. The authors should convince me (and their readers) that it is true. This should be carefully presented and discussed.
4- What is the effect of pressure and temperature gradients in the D-DIA column? Could you devise a strategy to identify stress such gradients?
5- line 179: "Our results show that in these two runs, the average stress estimated from various diffraction planes in pyrope is in the upper part of the stress values from different diffraction planes. This means that the strength of alumina and olivine aggregates are largely determined by the strong slip system(s) implying that deformation in these cases occur at nearly homogeneous strain."
I really do not understand this statement. The authors should extend and clarify their discussion.
6- Raterron et al, 2013, did run a EPSC model to evaluate the "true" stress in compressed alumina and compared their "true" stress to hkl-derived stresses as in Fig. 4 in the present paper. How do the results compare? Is the ordering of the hkl-stresses the same? How is the EPSC stress relative to that deduced from pyrope?
7- The resolution of the figures is terrible. This should be improved before publication.
Author Response
Response to Reviewer 3 Comments
Comments and Suggestions for Authors
Review for "Development of a stress sensor for in-situ high-pressure deformation experiments using radial X-ray diffraction" by J. Girard, R.E. Silber, A. Mohiuddin, H. Chen, and S. Karato, submitted to Minerals
This papers presents results regarding high pressure deformation experiments. The authors present deformation results on pyrope and argue that, because of its elastic and plastic anisotropy, garnet is a good material to serve as a reference for stress evaluation in diffraction experiments.
The main conclusions are new, scientifically sound, supported by the data, and useful to the community. As such, this paper should be published. I do have questions and comments (see below). I hence recommend publication after corrections.
1- Using stress sensors in D-DIA experiments is not new. Several groups used the alumina piston to evaluate stress in olivine, for instance. Scientific honesty requires citations to the relevant literature.
We would like to thank the reviewer for their thoughtful comments. We agree with the reviewer and we added the following to the introduction: “previous studies, especially when the sample itself couldn’t be used to estimate the stress during deformation (e.g. single crystal deformation) other polycrystalline materials such as alumina were used to estimate stress (e.g. [1-5] ). The stress sensor that was inserted into a sample space from which the average stress acting on a sample is estimated”; see lines (58-62). However, materials such as alumina have high plastic and elastic anisotropy and do not provide useful estimate of an average stress acting on a sample. In the present study, using a theory by Karato, we identify a good material as a stress sensor (pyrope) and report the experimental results, and discuss some implications for the nature of plastic deformation in alumina and olivine” (line 58-62)
2- Alumina as a stress sensor, how x-ray stresses could be corrected using EPSC models, and the effect of gradients in the D-DIA column were tested and published in Raterron et al Rev. Sci. Instrum. 2013. This paper should be compared to the present results and discussed.
We would like to thank the reviewer for this suggestion we added a comparison and a discussion to the main text Line 198-208.
3- Line 133: "When friction along the edge of each sample is small, stress in stress in each sample and in the stress sensor is same".
This typo is corrected. Thank you for bringing this to our attention.
Actually, no. If 3 is the vertical direction. s33 is, in fact, the same in the sample as well as in the stress tensor. Stress, as defined in Eq. 1 however is not s33, but sU = s33-s11. The authors hence assume that s11 is the same in all materials.
Is s11 the same in all materials? I do not know. The authors should convince me (and their readers) that it is true. This should be carefully presented and discussed.
As per the reviewer’s convention, s11 is related to the pressure in the cell and therefore it should be the same in the cell as far as a cell is made of relatively soft materials.
4- What is the effect of pressure and temperature gradients in the D-DIA column? Could you devise a strategy to identify stress such gradients?
Pressure in a sample assembly in a large volume high-pressure apparatus is usually homogeneous as far as sample assembly is made of soft materials. However, large temperature gradients are possible due to the furnace design. This was shown by Raterron et al (2013) with a simple graphite sleeve furnace 150C/mm gradient can be observed. Of course, temperature gradient would lead to problem in pressure estimate and uncertainty in stress estimate. To minimize the temperature gradient, we used a step heater as well as a straight heater that has been used in most previous studies. Using finite element thermal modeling we design a step furnace that drastically reduce the temperature gradient in the cell with only 50C different through the sample. We used this furnace design (figure 1b) in the run San 452. Stresses estimated in pyrope in the center and the top of the cell were very similar.
5- line 179: "Our results show that in these two runs, the average stress estimated from various diffraction planes in pyrope is in the upper part of the stress values from different diffraction planes. This means that the strength of alumina and olivine aggregates are largely determined by the strong slip system(s) implying that deformation in these cases occur at nearly homogeneous strain."
I really do not understand this statement. The authors should extend and clarify their discussion.
We tried to make this sentence clearer. “Our results show that in these two runs, the average stress estimated from various diffraction planes in pyrope (represented by the dotted and dash lines) is in the upper part of the stress values estimated from different diffraction planes in olivine and alumina. This means that the strength of alumina and olivine aggregates are largely determined by their respective strong slip system(s) (since the largest stress is required to deform the aggregate), and therefore implies that deformation of each aggregate in these cases occurs at nearly homogeneous strain.”
6- Raterron et al, 2013, did run a EPSC model to evaluate the "true" stress in compressed alumina and compared their "true" stress to hkl-derived stresses as in Fig. 4 in the present paper. How do the results compare? Is the ordering of the hkl-stresses the same? How is the EPSC stress relative to that deduced from pyrope?
We would like to thank the reviewer for this comment. We agree and added a paragraph to discuss this topic in the results and discussion section (line 198-208)
“Finally, we compared the stress estimated in alumina from our study with Raterron et al., [6]. We confirm that the ordering of the is the same in both studies (e.g., stress estimated from (012) diffraction plane is always the largest and (116) diffraction plane always show the smallest stress). However, in Raterron et al., [6], large anisotropy in stress estimated from alumina prevented an accurate stress estimation. Thus, they use EPSC modeling to calculate “true” stress. Their results show that at the end of deformation, EPSC modeling agrees with the lower end of stresses estimated in alumina. This is not consistent with our observations using a pyrope stress sensor. Since, those two methods are different, it is difficult to explain where the differences come from. In our study, stress in pyrope is directly measured from X-ray diffraction but the “true” stress corresponding to the EPSC modeling was calculating following some assumptions. Therefore, some of the EPSC assumptions might need to be refined.”
7- The resolution of the figures is terrible. This should be improved before publication.
We believe that is the issue of formatting. However, the figures were edited for a better resolution.
Girard, J.; Chen, J.; Raterron, P.; Holyoke, C.W. Deformation of periclase single crystals at high pressure and high temperature: Quantification of the effect of pressure on slip-system activities. Journal of Applied physics 2012, 111, 112607-112607-112605. Girard, J.; Chen, J.; Raterron, P.; Holyoke, C.W. Hydrolytic weakening of olivine at mantle pressure: Evidence of [100](010) slip system softening from single-crystal deformation experiements. Physics of the Earth and Planetary Interiors 2013, 216, 12-20. Raterron, P.; Amiguet, E.; Chen, J.; Li, L.; Cordier, P. Experimental deformation of olivine single crystals at mantle pressures and temperatures. Physics of the Earth and Planetary Interiors 2009, 172, 74-83. Raterron, P.; Chen, J.; Li, L.; Weidner, D.J.; Cordier, P. Pressure-induced slip-system transition in forsterite: Single crystal rheological properties at mantle pressure and temperature. American Mineralogist 2007, 92, 1436-1445. Amiguet, E.; Raterron, P.; Cordier, P.; Couvy, H.; Chen, J. Deformation of diopside single crystal at mantle pressure 1: Mechanical data. Physics of Earth and Planetary Interiors 2009, 177, 122-129. Raterron, P.; Merkel, S.; Holyoke, C.W. Axial temperature gradient and stress measurements in the deformation-dia cell, using alumina pistons. Review of Scientific Instruments 2013, 84.
